# Fuzzy PyTorch: Rapid Numerical Variability Evaluation for Deep Learning Models

**Inés Gonzalez-Pepe**                                                    *i_gon@live.concordia.ca*
*Department of Computer Science and Software Engineering*
*Concordia University*
*Montreal, Canada*

**Hiba Akhaddar**                                                        *h_akhadd@live.concordia.ca*
*Department of Computer Science and Software Engineering*
*Concordia University*
*Montreal, Canada*

**Tristan Glatard**                                                         *tristan.glatard@camh.ca*
*Krembil Centre for Neuroinformatics*
*Centre for Addiction and Mental Health*
*Toronto, Canada*

**Yohan Chatelain**                                                      *yohan.chatelain@camh.com*
*Krembil Centre for Neuroinformatics*
*Centre for Addiction and Mental Health*
*Toronto, Canada*

**Reviewed on OpenReview:** *https://openreview.net/forum?id=Oogq232VGP&noteId=BtOp1tw1dN*

## Abstract

We introduce Fuzzy PyTorch, a framework for rapid evaluation of numerical variability in deep learning (DL) models. As DL is increasingly applied to diverse tasks, understanding variability from floating-point arithmetic is essential to ensure robust and reliable performance. Tools assessing such variability must be scalable, efficient, and integrate seamlessly with existing frameworks while minimizing code modifications. Fuzzy PyTorch enables this by integrating stochastic arithmetic into PyTorch through Probabilistic Rounding with Instruction Set Management, a novel library interfacing with Verificarlo, a numerical analysis compiler. The library offers stochastic rounding mode and a novel mode; up-down rounding. Comparative evaluations show Fuzzy PyTorch maintains model performance and achieves runtime reductions of $5\times$ to $60\times$ versus Verrou, a state-of-the-art tool. We further demonstrate scalability by running models from 1 to 341 million parameters, confirming applicability across small and large DL architectures. Overall, Fuzzy PyTorch provides an efficient, scalable, and practical solution for assessing numerical variability in deep learning, enabling researchers and practitioners to quantify and manage floating-point uncertainty without compromising performance or computational efficiency.

## 1 Introduction

Many scientific domains have increasingly adopted deep learning (DL) models for computational analysis. However, these models often rely on numerical computations that are sensitive to variations in floating-point precision and rounding modes. Understanding numerical variability in DL models is therefore essential for ensuring reliable and reproducible outcomes. This is particularly critical in high-stakes applications such as medical imaging (Des Ligneris et al., 2023; Gonzalez-Pepe et al., 2023; Vila et al., 2024), remote sensing (Vicuna et al., 2021), and scientific simulations (Chatelain et al., 2022). Meanwhile, the importance

of numerical variability is beginning to gain industry-wide recognition. For example, variability analysis is now supported in Amazon Web Services' (AWS) Neuron SDK (AWS Neuron Team, n.d.), AWS Trainium chips, AMD MI300 GPUs, Tesla D1s and Blackwell architecture NVIDIA GPUs (El Arar, 2025). This reflects a growing demand for tools and frameworks that enable low-level control over numerical behavior in AI systems.

Although exact methods such as interval arithmetic (Hickey et al., 2001), symbolic execution (Solovyev et al., 2018), and formal verification (Boldo & Melquiond, 2011) exist to evaluate numerical accuracy, these approaches often require extensive modifications to the codebase and do not scale efficiently to complex deep learning workloads. Instead, we adopt stochastic arithmetic, a technique that introduces controlled random perturbations to floating-point operations. This enables statistical estimation of the numerical variability without modifying the underlying model architecture, making it more practical for large-scale applications. Stochastic arithmetic encompasses various techniques, including Monte Carlo Arithmetic (MCA) (Parker, 1997), CESTAC (Brunet & Chatelin, 1986) and Stochastic Rounding (Forsythe, 1959). To leverage stochastic arithmetic, researchers have developed tools such as Verificarlo (Denis et al., 2016), Verrou (Févotte & Lathuiliere, 2016), the stochastic rounding library by Fasi and Mikaitis (Fasi & Mikaitis, 2021) and CADNA (Jézéquel & Chesneaux, 2008).

While stochastic arithmetic has been widely explored in numerical analysis, it remains underused in deep learning, despite recent studies demonstrating its potential to assess uncertainty in neural network training and inference (Faraone & Leong, 2019; Kloberdanz et al., 2022; Beuzeville et al., 2024; Arar et al., 2025). A major practical challenge of stochastic arithmetic is the need to run programs multiple times, typically 10 times or more, to obtain stable statistical estimates of their variability. However, currently available stochastic arithmetic tools, especially Verrou, one of the more accessible stochastic arithmetic frameworks for DL, introduce slowdowns of $10\times$ to $1000\times$ on DL models with only a few million parameters. In such settings, collecting enough samples takes days or weeks, even with parallelization, and scaling these analyses to modern large language models becomes effectively infeasible. For numerical variability research to keep pace with rapidly growing DL architectures, it is therefore essential to develop tools that introduce minimal computational overhead. This motivates the design goals of Fuzzy PyTorch, which prioritizes speed and scalability so that numerical variability studies remain tractable even on increasingly large models.

Fuzzy PyTorch integrates stochastic arithmetic, more specifically Monte Carlo Arithmetic (MCA), into the PyTorch framework through a novel library named **P**robabilistic **R**ounding with **I**nstruction **S**et **M**anagement (PRISM). PRISM implements two modes: stochastic rounding (SR) and Up-Down rounding (UD). SR bypasses exact operations and therefore preserves exact values by probabilistically rounding values based on their proximity to representable floating-point numbers. Meanwhile, UD is a newly proposed rounding mode that is faster at the level of individual operations, as it randomly rounds up or down with equal probability. Both modes are optimized using vectorized CPU instructions through the Highway library (Google, 2024a), minimizing computational overhead. Fuzzy PyTorch seamlessly integrates with Py-Torch by extending the Verificarlo compiler, providing a fast, practical framework for numerical variability analysis in deep learning. By integrating directly with the PyTorch execution pipeline and avoiding the major bottlenecks present in traditional approaches, such as the strict serialization in Verrou or the lack of vectorized support in CADNA and standard MCA, Fuzzy PyTorch enables full-scale analyses that were previously computationally prohibitive. Compared to state-of-the-art tools, Fuzzy PyTorch achieves similar numerical-error characteristics but with markedly lower runtime, allowing researchers to evaluate numerical variability across full DL workflows. This efficiency makes large-scale, systematic studies of floating-point behavior feasible, directly supporting research on reproducibility, robustness, and numerical uncertainty in modern neural networks and helping advance reproducible and principled DL research.

This work proposes three main contributions:

1. **PRISM:** We introduce PRISM, which implements fast probabilistic rounding methods for the systematic analysis of floating-point errors.

2. **Stochastic arithmetic in PyTorch:** We seamlessly integrate stochastic arithmetic into PyTorch, enabling efficient and transparent instrumentation of deep learning operations.

3. **Comparative evaluation with Verrou:** We perform a comprehensive evaluation against Verrou, a state-of-the-art tool for numerical variability analysis, showcasing the enhanced performance and flexibility of Fuzzy PyTorch on use cases ranging from digit classification with MNIST (LeCun, 1998), whole brain MRI segmentation with the FastSurfer neuroimaging model (Henschel et al., 2020), and Parkinson's classification from speech data with the WavLM model (Chen et al., 2022).

The remainder of this paper is structured as follows: Section 2 details the design and implementation of Fuzzy Pytorch, including the UD rounding mode, the PRISM library implementation and PyTorch instrumentation. Section 3 presents results validating numerical accuracy and demonstrating 5–60× runtime speedups over Verrou. Section 4 concludes with discussion of limitations and future directions. Supplementary material, including additional information on existing rounding modes, use cases, the algorithm for probabilistic rounding and further statistical analysis of the harmonic series, is provided in the Appendix.

## 2 Fuzzy Pytorch Design and Implementation

Fuzzy Pytorch implements a new rounding mode, Up-Down rounding (subsection 2.1), a faster alternative to stochastic rounding implemented in the PRISM library (subsection 2.2). The PRISM library implements the SR and UD rounding modes and the modifications to the Verificarlo compiler for compiling PyTorch with the PRISM library.

### 2.1 Up-Down Rounding

The Up-Down Rounding (UD) technique rounds the result of an already rounded floating-point operation to the next or previous floating-point number with equal probabilities. Although UD rounding does not preserve exact floating-point operations, it produces results similar to SR rounding on large code bases, while generally being significantly faster. UD rounding is defined as:

$$\circ_{\text{UD}}(x) = \begin{cases} \circ_{\text{RN}}(x) - \epsilon(x) & \text{with probability } \frac{1}{2} \\ \circ_{\text{RN}}(x) + \epsilon(x) & \text{with probability } \frac{1}{2} \end{cases} \tag{1}$$

where $\epsilon(x)$ is the *unit in the last place* (Muller et al., 2018) if $x \neq 0$ and 0 otherwise, and $\circ_{\text{RN}}(x)$ is the IEEE-754 round-to-nearest with ties-to-even rounding mode. Additional details and formal definitions of stochastic rounding, as well as the other rounding modes used for comparison to UD (CESTAC, and IEEE-754), are provided in Appendix A.

### 2.2 PRISM

PRISM is a `C++` library that implements the SR (Appendix A.2.2) and UD rounding (sub-section 2.1) modes. We do not currently plan to support CESTAC, as it is not as commonly used as SR, but we consider it as a future work direction. PRISM leverages the Highway library (Google, 2025) to use vectorized instructions available on modern CPUs, thereby minimizing the overhead introduced by stochastic arithmetic. Highway selects the best architecture target to generate the most efficient code for the CPU, either at compile time (static dispatch) or runtime (dynamic dispatch).

PRISM provides probabilistic rounding (SR and UD) for the floating-point operations $\{+, -, \div, \times, \sqrt{}, \text{Fused Multiply-Add (FMA)}\}$. The SR operators (except FMA) are implemented using the rounding-mode-free algorithms by (Fasi & Mikaitis, 2021). We extend these algorithms to support the FMA instruction (described in Algorithm 1). Our FMA implementation is inspired by Verrou and is based on the ErrFmaNearest Algorithm by (Boldo & Muller, 2010).

PRISM's interface offers functions for scalar and vector instructions, supporting both static and dynamic dispatch. The static dispatch versions accept vector types as inputs, while the dynamic dispatch versions accept pointers to scalar types. Dynamic dispatch is necessary because vector types may not be available at runtime (e.g., 512-bit AVX-512 registers on AVX2 architecture with 256-bit registers). Although slightly slower, dynamic dispatch enhances portability, enabling x86-64 binaries to run on any architecture.

---

**Algorithm 1:** FMA With Stochastic Rounding Without the Change of the Rounding Mode

---

1: **function** FMA2$(a \in \mathcal{F}, b \in \mathcal{F}, c \in \mathcal{F})$
2:     Compute $\varrho = \circ_{\mathrm{SR}}(a \cdot b + c) \in \mathcal{F}$
3:     $Z \leftarrow \mathrm{rand}()$
4:     $\sigma \leftarrow \circ_{\mathrm{RN}}(a \cdot b + c)$
5:     $(u_1, u_2) \leftarrow \mathrm{TwoProdFMA}(a, b)$
6:     $(\alpha_1, \alpha_2) \leftarrow \mathrm{TwoSum}(c, u_2)$
7:     $(\beta_1, \beta_2) \leftarrow \mathrm{TwoSum}(u_1, \alpha_1)$
8:     $\gamma \leftarrow \circ_{\mathrm{RN}}(\circ_{\mathrm{RN}}(\beta_1 - r_1) - \beta_2)$
9:     $\tau \leftarrow \circ_{\mathrm{RN}}(\gamma + \alpha_2)$
10:    round $\leftarrow \mathrm{SRround}(\sigma, \tau, Z)$
11:    $\varrho \leftarrow \circ_{\mathrm{RN}}(r_1 + \mathrm{round})$
12:    **Return** result
13: **end function**

---

Finally, PRISM supports multithreaded execution by assigning a separate random generator to each thread, ensuring that concurrent executions do not share the same seed state. This enables optimal performance without requiring any synchronization mechanism and prevents correlations in the generated floating-point perturbations across threads.

We modified the Verificarlo compiler (Denis et al., 2016) to use the PRISM library. Verificarlo replaces floating-point operations with generic calls to configurable backends (e.g., MCA, IEEE, VPREC (Chatelain et al., 2019)) at the LLVM (Lattner & Adve, 2004) Intermediate Representation (IR) level. In its current version, Verificarlo serializes the vectorized instructions, which can cause additional slowdowns. It adheres to the Interflop (Defour et al., 2021) interface, which exposes scalar arithmetic operations but not vectorized ones. Specifically, we implemented new LLVM instrumentation passes to replace scalar and vectorized floating-point operations with calls to the PRISM library. We also ensured ABI compatibility between the PRISM library and the source code to prevent incorrect register use during argument passing.

## 2.3 PyTorch Instrumentation

We instrumented PyTorch version 2.2.1 with Verificarlo 2.0.0, using the PRISM library as the backend. Verificarlo was built with LLVM version 7.0.0, including support for FORTRAN code through LLVM's flang compiler. We used Python 3.8.5. To ensure compatibility with Verificarlo, we modified only one line in the PyTorch codebase. Specifically, we removed the `noexcept` keyword from the move constructor of the `Module` class in `torch/csrc/jit/api/module.h`. This adjustment was necessary to prevent compilation errors related to LLVM compatibility but should no longer be required with more recent LLVM versions.

To achieve complete instrumentation, we compiled the open-source BLAS and LAPACK libraries (Anderson et al., 1999) with Verificarlo, replacing the proprietary Intel MKL library as the default BLAS implementation. Architecture-specific instructions (`-march=native`) were enabled to leverage vectorized operations. Additionally, the ONNX (Community, 2024) runtime was compiled with the PRISM library to ensure comprehensive instrumentation of the entire model execution. We disabled the use of the Intel MKL DNN (Corporation, 2024) library to avoid reliance on proprietary software. We did not instrument the protobuf (Google, 2024b) third-party library to avoid perturbing model serialization. We conducted the experiments using the software versions listed above. While we have also instrumented PyTorch 2.6.0 with Python 3.10 and LLVM 11.0.0 via Verificarlo 2.2.0 (the code is available for compilation in our documentation), these were not used in the experiments reported in this paper.

We excluded specific functions from instrumentation because they were susceptible to producing erroneous outputs under our rounding modes. Correct instrumentation alternatives are under development. In particular, PyTorch's exponential and logarithmic functions in the SLEEF (SIMD Library for Evaluating Elementary Functions) third-party library exhibited large output deviations when perturbed, especially when input values approached zero. Similarly, the `torch.argmax` operation became unreliable under UD rounding, likely

because approximate rounding can alter comparisons when input values are close, affecting the selected index. As these functions are sensitive and integral to correct model behavior, they were not instrumented to ensure the reliability of the current results.

### 2.4 Availability of Data and Code

The code and data used in this study are available on GitHub at `https://github.com/big-data-lab-team/fuzzy-pytorch`. The repository includes the PRISM library, the modified Verificarlo compiler, the Dockerfile to build Fuzzy PyTorch, and the scripts used to run the experiments.

### 2.5 Computational Infrastructure

All experiments except for the WavLM experiment were conducted on a server equipped with 8 compute nodes with 32 cores Intel(R) Xeon(R) Gold 6130 CPU @ 2.10GHz 22MB cache L3. The WavLM experiment was conducted on the Narval cluster from École de Technologie Supérieure (ETS, Montréal), managed by Calcul Québec and The Digital Alliance of Canada which includes AMD Rome 7502, AMD Rome 7532, and AMD Milan 7413 CPUs with 48 to 64 physical cores, 249 GB to 4000 GB of RAM and Linux kernel 3.10.

## 3 Results

We evaluated the accuracy, runtime efficiency, and numerical variability of Fuzzy PyTorch using Verrou, CADNA, and the stochastic rounding library of Fasi and Mikaitis as baselines. We performed (1) a sanity check with the harmonic series, a classical example in numerical analysis, (2) a computational overhead assessment with the NAS Parallel Benchmarks, demonstrating Fuzzy PyTorch's performance in an HPC context, and (3) runtime performance and numerical variability assessments in three practical deep-learning applications: MNIST digit classification, FastSurfer CNN brain segmentation, and WavLM-based speech classification.

### 3.1 Numerical Error Estimation Baselines

We compared the PRISM library with several state-of-the-art tools for numerical variability analysis. Verrou (Nethercote & Seward, 2007) is a Valgrind-based Dynamic Binary Instrumentation tool that perturbs floating-point operations by replacing them with custom rounding functions, supporting modes such as Stochastic Rounding (SR, called average rounding) and asynchronous CESTAC (called random rounding), and can instrument binaries without recompilation, though multithreading is serialized due to Valgrind. The Verificarlo MCA backend (Parker, 1997) implements Monte Carlo Arithmetic by injecting uniform perturbations with a user-defined virtual precision $t$, performing computations at twice the target precision to avoid double rounding, and offering modes including Random Rounding (equivalent[1] to SR when $t = p$), but incurs overhead from serializing vectorized instructions and using 128-bit arithmetic for binary64. CADNA (Jézéquel & Chesneaux, 2008) provides a CESTAC-based synchronous stochastic arithmetic library via overloaded stochastic types that propagate three perturbed values per operation and estimate significance loss (e.g., "computational zero"), with OpenMP support for threads but no vectorized instruction support. Finally, the stochastic rounding library of Fasi and Mikaitis (FM) (Fasi & Mikaitis, 2021) offers C functions implementing SR for scalar floating-point operations (excluding FMA), requiring explicit replacement of each arithmetic operation and lacking vectorized interfaces. We will refer to this library as FM SR in the remainder of the paper.

### 3.2 Harmonic Series Validates Expected Variability Patterns

To evaluate the accuracy of the UD and SR rounding modes, we analyzed the harmonic series $\sum_{i=1}^{n} \frac{1}{i}$ for $n$ ranging from $10^2$ to $10^7$. While this series is divergent in real arithmetic, it converges in floating-point arithmetic for $n \geq 2^{48}$ within IEEE-754 binary32 format due to numerical absorption (Malone, 2013). We

---

[1]The MCA RR mode is biased for inputs close to a power of two, see (de Oliveira Castro, 2022).

performed computations in IEEE-754 binary32 format and compared PRISM against established baselines: CESTAC (via CADNA), MCA RR (via Verificarlo), Verrou (SR and CESTAC modes) and FM SR. A binary64 computation served as the reference value.

Figure 1 presents the standard deviation and mean values obtained from three repetitions per mode (the three internal values extracted from a single run for CADNA). The results show that PRISM SR exhibits variability comparable to MCA RR, Verrou SR, and FM SR (Levene's test, $p = 0.29$; see Extended Data Table 2). In contrast, UD rounding displays higher variability, which is expected as it applies perturbations to values that have already been rounded. Finally, CADNA exhibits the highest variability as it ensures different rounding directions across its three internal repetitions.

As shown in Figure 1, the mean values exhibit three distinct behaviors:

1. The MCA SR, Verrou SR, PRISM SR, and FM SR modes closely approximate the binary64 reference value. This aligns with expectations, as SR rounding converges to the expected value.

2. The CESTAC and Verrou CESTAC modes diverge more rapidly. This behavior is consistent with the known bias introduced by CESTAC rounding.

3. The PRISM UD mode converges toward the binary32 result, as expected, since UD rounding applies random perturbations to values that have already been rounded using the round-to-nearest mode.

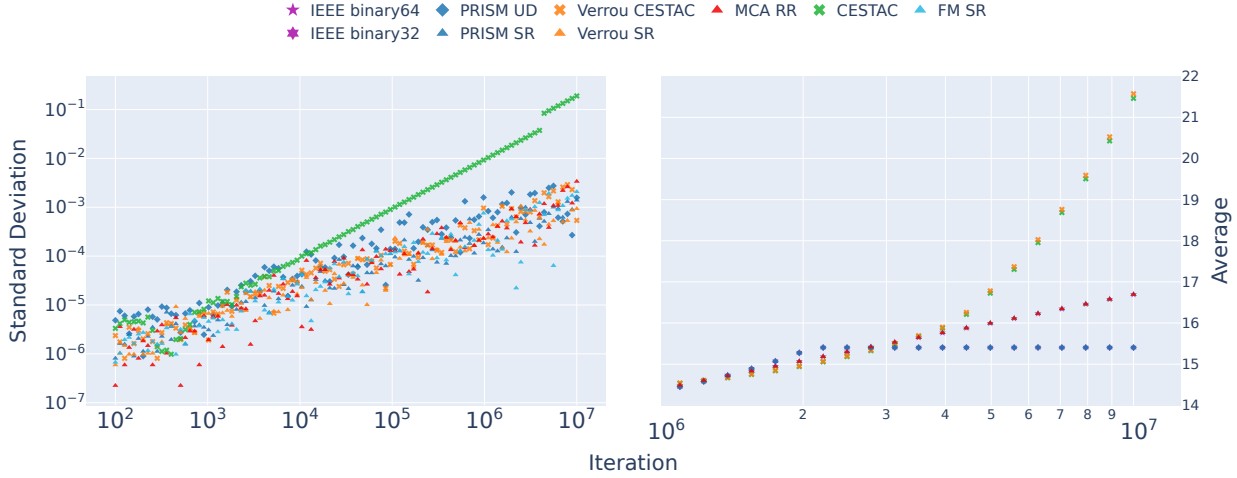

Figure 1: Comparison of probabilistic rounding on the harmonic series example.

### 3.3 PRISM UD Minimizes Runtime Slowdowns for NAS Parallel Benchmarks

We evaluated the runtime overhead introduced by PRISM using the `C++` NAS Parallel Benchmarks version 3.4.1 (Löff et al., 2021) (Appendix Table 1). The Integer Sort (IS) benchmark was excluded, as it does not involve floating-point operations.

Experiments were performed using the serial implementation on dataset classes $S$ and $A$, corresponding to the smallest benchmark workloads. PRISM SR and UD modes were compared against CESTAC (via CADNA), MCA RR (via Verificarlo), Verrou (SR and CESTAC modes) and FM SR. All benchmarks were compiled with `-march=haswell -maes` to enable AVX2 instructions and executed using IEEE-754 binary64 arithmetic. Runtimes were averaged over three independent repetitions. An uninstrumented execution was used as the baseline.

Figure 2 shows that PRISM SR induces runtime slowdowns that are comparable to, or lower than, those observed for Verrou SR and CADNA. PRISM UD consistently yields the lowest overhead among the evaluated tools. These trends are consistent across dataset classes.

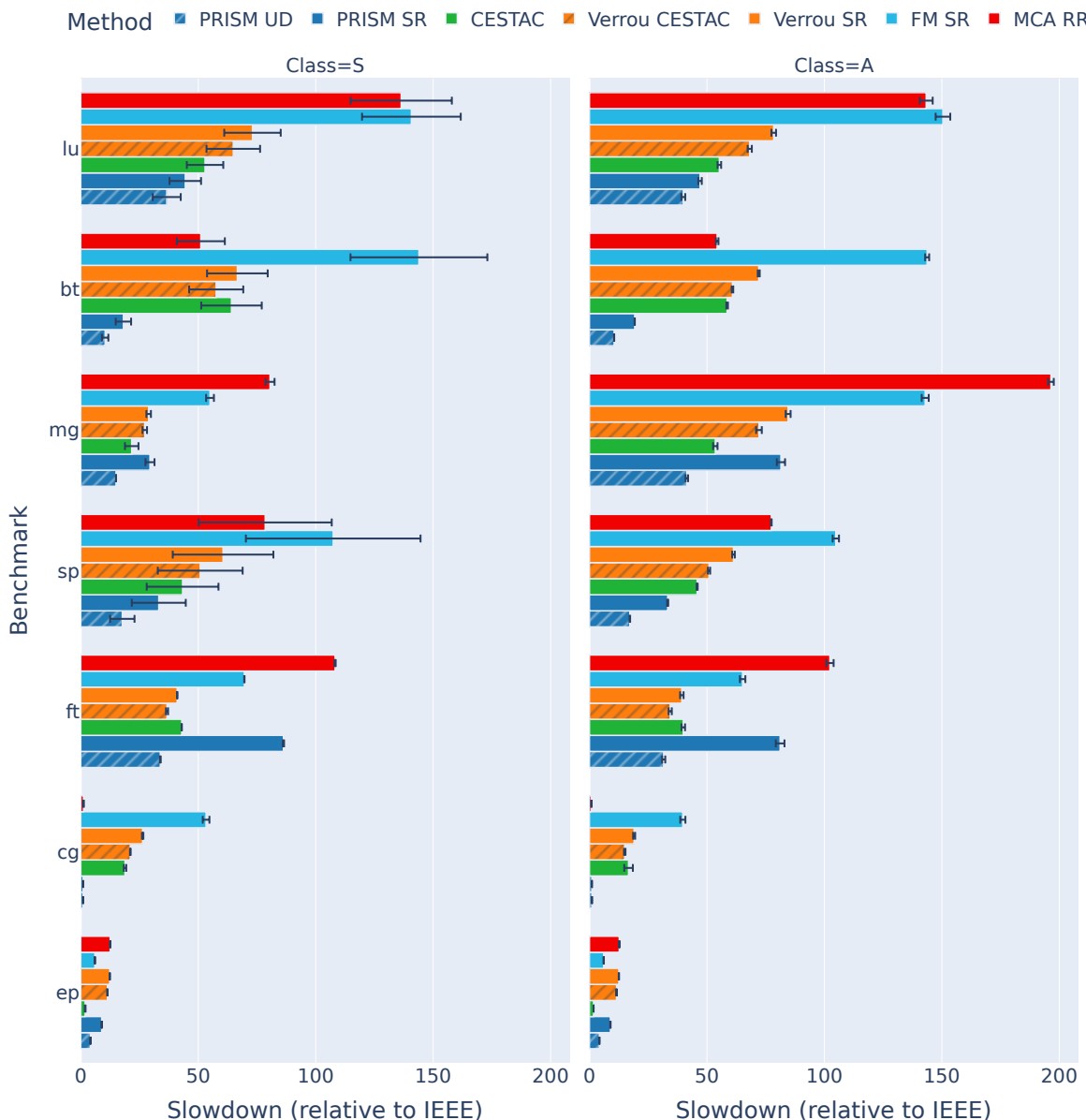

Figure 2: Comparison of slowdowns across numerical variability analysis tools for NAS Parallel Benchmarks on dataset S and A, relative to the IEEE binary64 baseline.

### 3.4 Fuzzy PyTorch Achieves Significant Runtime Speedup

To further evaluate Fuzzy PyTorch's efficiency in practical DL workflows, we measured inference runtime for MNIST, FastSurfer and WavLM (see model descriptions in Appendix C). We compared PRISM UD and SR modes against Verrou's CESTAC and SR modes. All experiments were executed using a single thread, as Verrou enforces serialization of multithreaded execution.

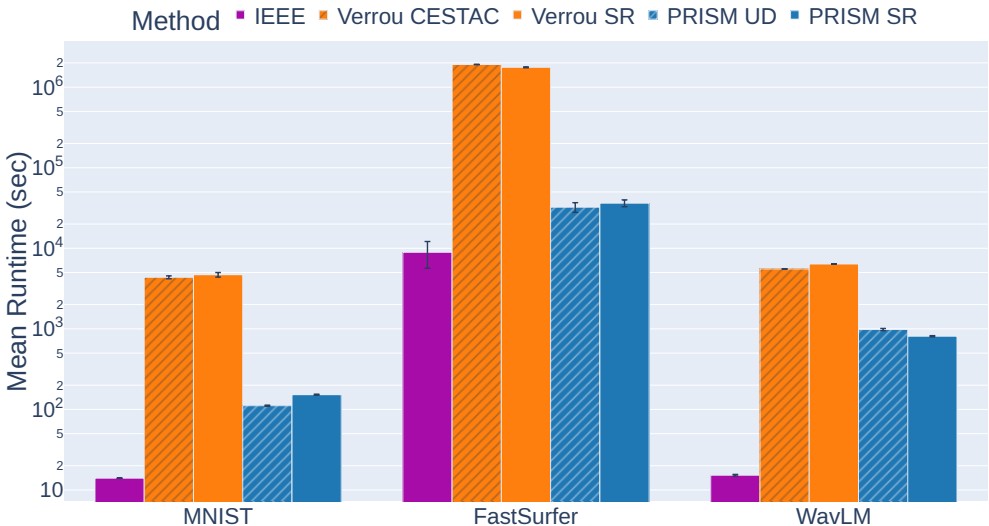

Figure 3: Comparison of instrumentation runtimes across DL models

As seen in Figure 3, Fuzzy PyTorch achieves speedups ranging from 5× to 60× for UD mode and 7× to 49× for SR mode compared to Verrou. This improvement is notable considering that Verrou's PyTorch instrumentation relies on the highly optimized Intel MKL library, whereas Fuzzy PyTorch uses the standard Netlib BLAS/LAPACK implementation.

In contrast to the NAS Parallel Benchmarks, where PRISM UD consistently outperformed PRISM SR, the relative performance of UD and SR varies across deep learning workloads. For WavLM, SR achieves a larger speedup (7.89× relative to Verrou) than UD (5.65×). UD retains an advantage for the CNN-based architectures FastSurfer (UD: 60.43×, SR: 49.07×) and MNIST (UD: 39.22×, SR: 30.81×).

We attribute the UD slowdown on WavLM to its transformer-based architecture. Transformer models are computationally more complex than CNNs: self-attention layers introduce quadratic cost in sequence length with heavier intermediate memory usage. Unlike UD which incurs a constant perturbation cost for every operation, SR bypasses exact operations, reducing its relative overhead.

### 3.5 Comparable Numerical Variability between Fuzzy PyTorch and Verrou

To assess Fuzzy PyTorch beyond runtime performance, we evaluated the numerical variability it introduces compared to Verrou across MNIST, FastSurfer and WavLM.

For MNIST in Figure 4a, we evaluated standard classification metrics, including accuracy and weighted precision, recall, and F1 score, and quantified numerical variability using the significant digits metric (Sohier et al., 2021). In binary32 arithmetic, the theoretical upper bound is 7.23 significant digits; across all metrics we observed a maximum of approximately 6.17 significant digits, with the loss function exhibiting noticeably higher variability. Variability is effectively confined to the loss across both Fuzzy PyTorch and Verrou, which is expected given that MNIST is a well-solved task with highly stable predictions. UD rounding consistently introduces greater variability than stochastic rounding for both tools. Fuzzy PyTorch shows slightly lower significant digits overall, which we attribute to its instrumentation of AVX-512 vector instructions, allowing a wider class of floating-point operations to be perturbed compared to Verrou.

In the WavLM use case (Figure 4b), as with MNIST, we evaluated numerical variability across accuracy and the weighted variants of precision, recall, and F1 score. On average, we observe 6.17 significant digits across all performance metrics, indicating high numerical stability—comparable to that observed across IEEE executions.

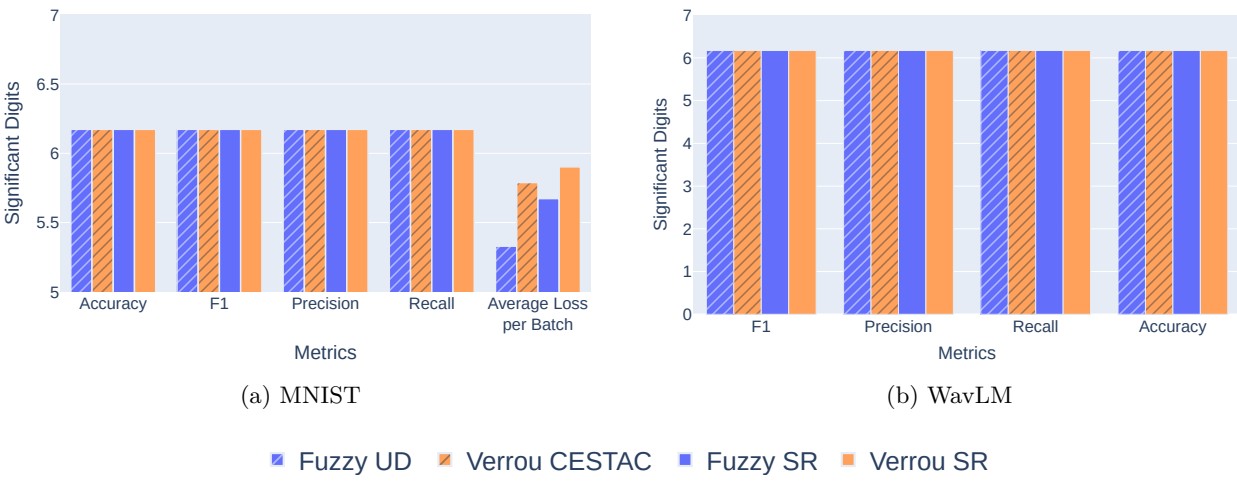

(a) MNIST

(b) WavLM

▨ Fuzzy UD    ▨ Verrou CESTAC    ▪ Fuzzy SR    ▪ Verrou SR

Figure 4: Significant digits across MNIST and WavLM model metrics for different instrumentation tools.

To investigate whether this apparent stability conceals underlying instability, we analyzed the model's output probabilities before the final max operation. As shown in Figure 5, the number of significant digits drops, averaging around 4 across all modes and tools, with a standard deviation of approximately half a digit. This suggests that some numerical instability is indeed present but is masked by the final max operation. Consistent with findings from previous use cases, we also note that PRISM UD mode introduces the highest level of numerical perturbation—even surpassing Verrou's CESTAC mode.

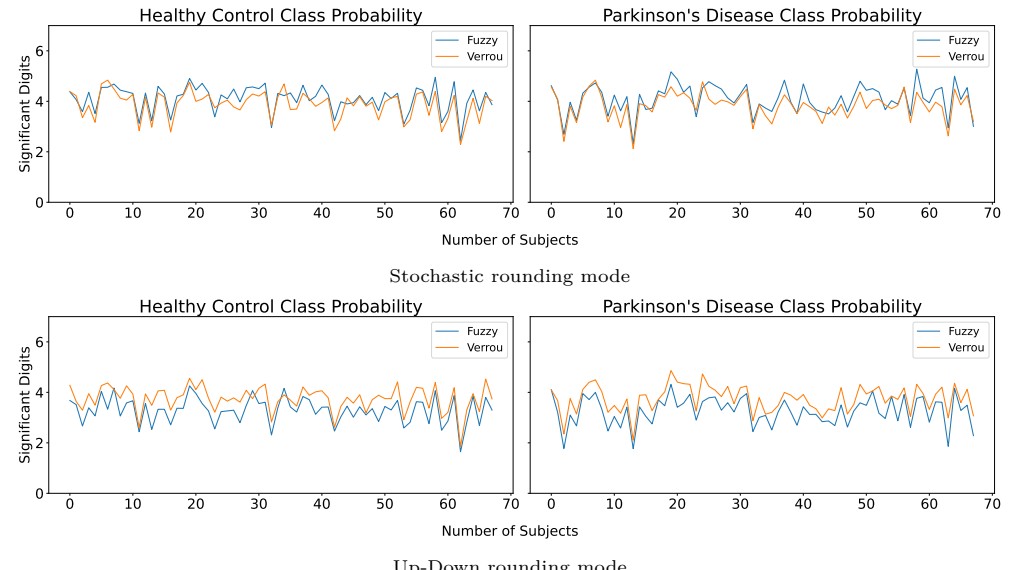

Figure 5: Significant digits for rounding modes across WavLM model's class probabilities

For the FastSurfer use case, we assess variability at inference using the minimum Sørensen–Dice scores between MCA iterations (Figure 6). The minimum Sørensen–Dice score captures the most extreme cases of variability across brain regions, offering a global measure of segmentation consistency. Across all modes, coefficients remain extremely high, with the lowest observed value approaching 0.9985. Variability magnitudes are similar across methods, though slightly higher for PRISM UD, likely due to its lack of preservation of exact operations. Similarly to the MNIST results, PRISM UD shows the highest variability but still main-

tains comparable segmentation accuracy. These results confirm that our MCA-based instrumentation for FastSurfer operates correctly, producing consistent and interpretable variability measurements across modes.

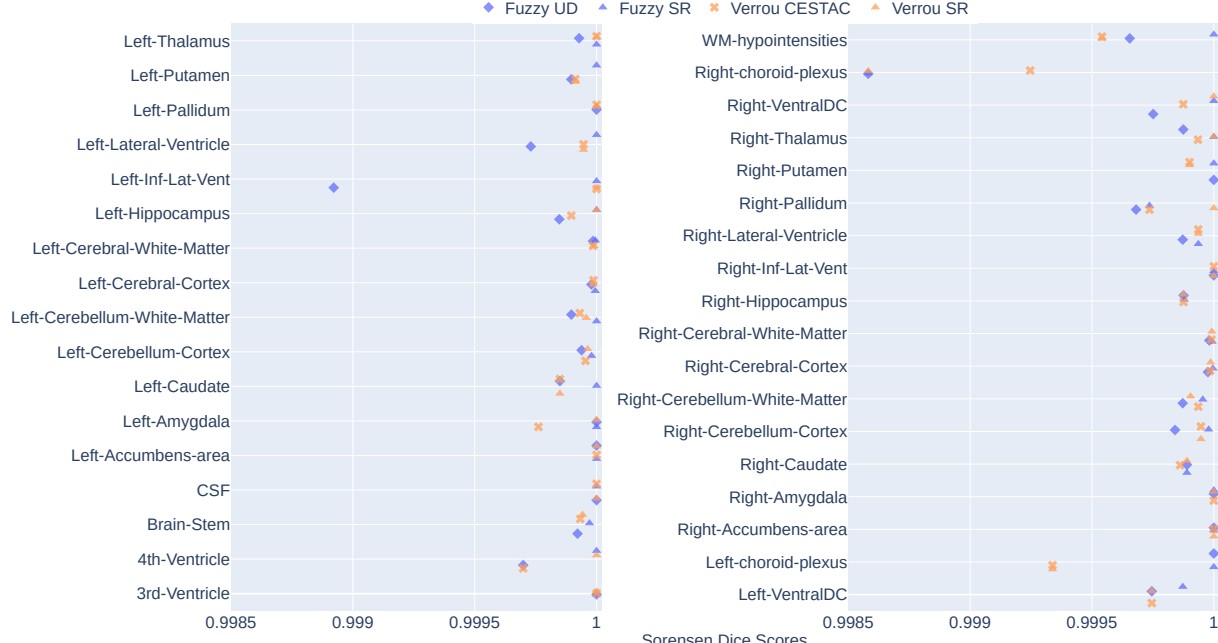

Figure 6: Minimum Sørensen-Dice score across instrumentation tools and different labelled brain regions.

For all use cases, we verified that no random processes were present after fixing the random seeds by running multiple iterations of the IEEE implementations of each model. In FastSurfer, this yielded a perfect minimum Sørensen–Dice score (1.0, standard deviation 0), confirming determinism. For MNIST and WavLM, all metrics, including loss, retained 6.17 significant digits.

## 3.6 Model Embeddings Are Comparable

To verify that numerical perturbations propagate correctly through the model architectures, we analyzed the variability of internal feature maps (embeddings). Figure 7 illustrates one embedding from each use case: the output of the second decoder block for FastSurfer, the first convolutional layer for MNIST and the output of the ECAPA-TDNN transformer component of the WavLM model.

Numerical variability followed consistent, task-dependent patterns across all tools and rounding modes, although, for WavLM, we can visibly see the slightly higher numerical uncertainty with Fuzzy PyTorch's UD mode compared to the other modes.

While the numerical uncertainty within the MNIST and WavLM embeddings aligns with the observed stability of its outputs, FastSurfer presented a notable discrepancy: substantial uncertainty in the background of its embeddings despite its stable outputs. Further investigation uncovered that this instability was confined to the background region outside the brain and originated from unstable indices being generated by the max-pooling operation across stochastic arithmetic iterations. Post-processing steps in FastSurfer effectively masked the background, thereby mitigating the impact of these instabilities on the final outputs for this specific use case. This finding directly motivated the development of Conservative & Aggressive NaNs, two approaches for leveraging numerical uncertainty into computational efficiency gains while preserving model performance (Gonzalez-Pepe et al., 2026).

Overall, these findings demonstrate consistency between the global patterns observed in Fuzzy PyTorch and Verrou's results, reinforcing the reliability of Fuzzy PyTorch in assessing variability throughout DL models.

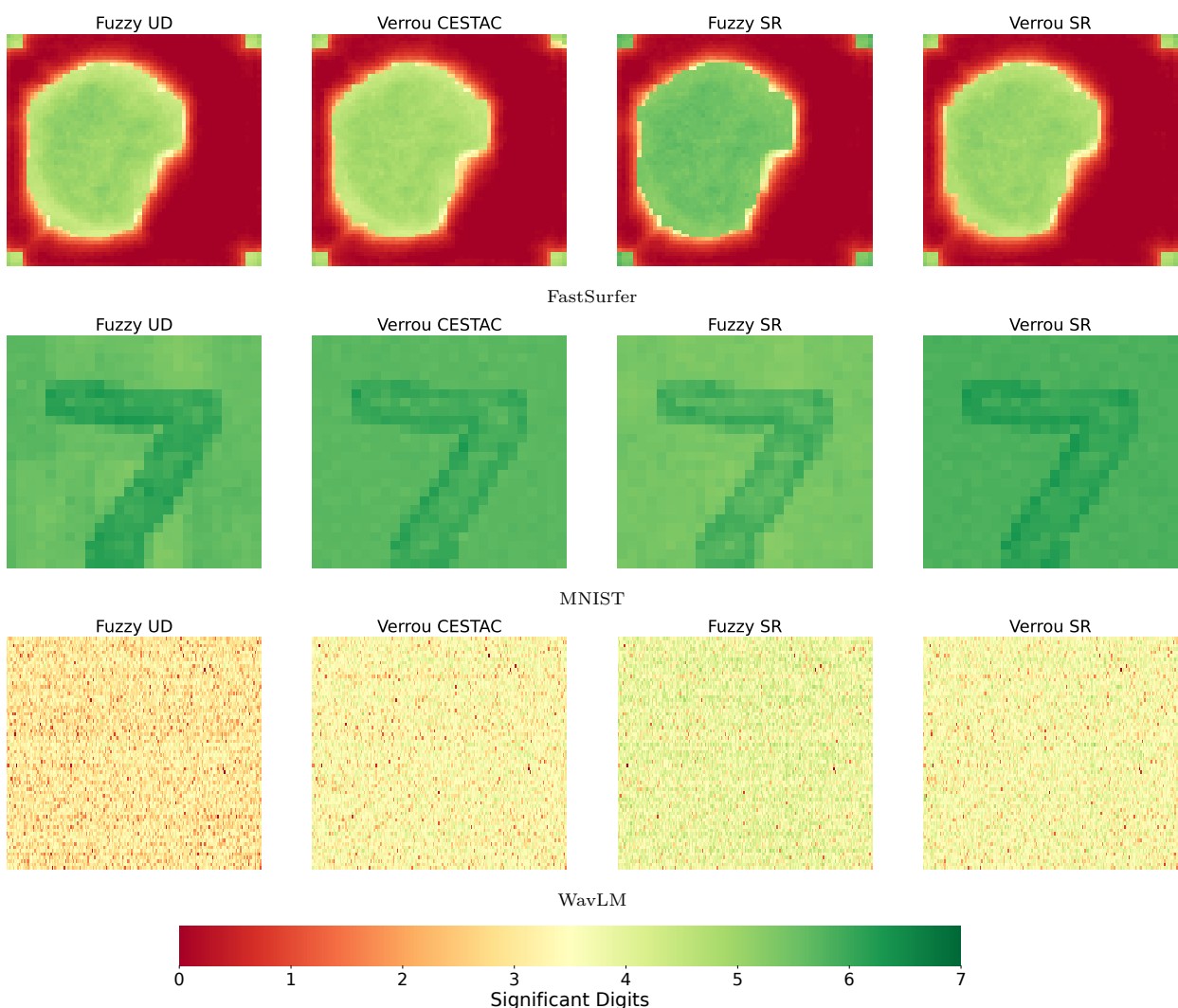

Figure 7: Model embeddings across stochastic arithmetic implementations

## 4 Conclusion

Fuzzy PyTorch is a framework designed to evaluate numerical variability in DL models, addressing the challenges posed by floating-point arithmetic limitations. By leveraging vectorized CPU instructions via the Highway library, it minimizes computational overhead while maintaining flexibility. The framework supports two rounding modes, SR and UD, allowing researchers to balance precision and computational efficiency, making it a versatile tool for enhancing model robustness and reproducibility. Across harmonic series tests and deep learning tasks, SR consistently provides the most accurate and stable results, while UD drifts toward lower precision and CESTAC diverges due to bias. Fuzzy PyTorch reproduces these variability patterns and achieves accuracy comparable to state-of-the-art tools such as Verrou across MNIST classification, FastSurfer segmentation, and WavLM Parkinson's detection.

Fuzzy PyTorch crucially achieves substantial speedups, with minimal slowdowns for PRISM SR and especially PRISM UD in NAS parallel benchmarks. In deep learning tasks, it reaches up to $60\times$ acceleration—despite using less optimized CPU libraries (OpenBLAS and LAPACK versus Intel MKL). While PRISM UD is fastest in smaller benchmarks, its advantage decreases in DL experiments, likely because it does not skip exact operations. Importantly, such speed is not only beneficial for small numerical tests but becomes essential in large-scale DL workloads, where evaluating numerical variability can otherwise be prohibitively slow. The ability to run multiple stochastic executions at scale is critical for feasible, timely numerical analysis. This efficiency gain enables the scalable, systematic evaluation of numerical variability in large DL models, a capability unmatched by existing methods.

As numerical variability becomes increasingly recognized across the AI industry, the availability of system-level controls—such as AWS Neuron's hardware-supported rounding modes—highlights the pressing need for software frameworks like Fuzzy PyTorch. These frameworks make it possible to perform fine-grained evaluations of numerical behavior across diverse architectures, from CPUs to accelerators, and ensure that both academic and industrial workflows benefit from deeper guarantees of computational robustness. By systematically analyzing variability, we can identify sources of instability that can be leveraged for practical improvements. For example, during our work with the FastSurfer CNN, Fuzzy PyTorch revealed regions of numerical instability that we subsequently exploited to implement Conservative & Aggressive NaNs, yielding significant computational efficiency gains (Gonzalez-Pepe et al., 2026). This illustrates how efficient variability analysis not only ensures reproducibility but also enables optimization and the development of novel techniques that improve performance in large-scale deep learning models.

While our current implementation is CPU-based, the main conclusions from CPU testing are expected to generalize to GPU architectures. Future work, involves extending the framework to GPU architectures, although it poses significant technical challenges. The GPU tooling ecosystem is far more fragmented than its CPU counterpart, requiring specialized approaches for different vendor platforms. A successful GPU implementation would need to address multiple compilation targets and runtime environments. For NVIDIA GPUs, dynamic instrumentation frameworks like NVBit (Villa et al., 2019) could enable analysis of closed-source libraries, but this approach is complex and platform-specific. Alternatively, compiler-based solutions using IREE (Liu et al., 2022) with specific compilation targets could offer broader hardware support, though this would require substantial development effort to integrate with existing PyTorch workflows. Nonetheless, we expect to make the Fuzzy PyTorch framework applicable to GPU-based deep learning workloads once Verificarlo extends its support to GPU architectures. Future directions for Fuzzy PyTorch, beyond GPU support, include exploring the impact of numerical variability on diverse DL model architectures, optimizing SR mode performance, extending PRISM with additional floating-point formats and specialized DL instructions.

In summary, Fuzzy PyTorch provides an efficient, reliable, and versatile tool for assessing numerical variability in DL models. It empowers researchers to deepen their understanding of numerical behavior in DL models and enhances their ability to develop robust and reproducible systems.

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

# A   Numerical Variability Estimation

Measuring numerical variability in DL models can involve a family of techniques that introduce randomness into floating-point computations. These methods rely on non-deterministic rounding. Unlike standard IEEE-754 rounding modes, this approach rounds to either of the two closest floating-point numbers based on computed probabilities. The specific stochastic arithmetic technique is determined by how this probability is computed. We use the term Probabilistic Rounding (PR), introduced in (Higham & Mary, 2019), as an umbrella term for this class of techniques. Moreover, because IEEE rounding is deterministic rather than probabilistic, we present it in its own subsection to emphasize this difference.

## A.1   IEEE-754 Rounding

The IEEE-754 round-to-nearest mode, also known as round-to-nearest, ties-to-even, is the default rounding mode used in most floating-point hardware and software implementations. When a real number cannot be exactly represented in floating point, it is rounded to the closest representable number. If the number lies exactly halfway between two floating-point values, the tie is broken by rounding to the one with an even least significant bit (i.e., the one whose mantissa ends in 0).

Formally, the rounding function $\circ_{\mathrm{RN}}(x)$ maps $x \in \mathbb{R}$ to the nearest floating-point number $f \in \mathcal{F}$ such that:

$$\circ_{\mathrm{RN}}(x) = \arg \min_{f \in \mathcal{F}} |x - f|$$

with ties resolved to the floating-point number $f$ such that the significand of $f$ is even. This method minimizes rounding bias over repeated operations and is the most widely adopted deterministic rounding strategy defined by the IEEE-754 standard (IEEE Computer Society, 2008), which specifies several modes including round-to-nearest (tie-to-even or tie-to-odd), rounding toward $\pm\infty$, and rounding toward zero.

## A.2   Probabilistic Rounding

Let $\mathcal{F}$ be the set of normalized binary floating-point numbers with elements $x = (-1)^s.m.2^e$, where $s \in \{0, 1\}$ is the sign bit, $2^{p-1} \leq m < 2^p$ is the significand, $e \in \mathbb{Z}$ is the exponent and $p$ is the precision. Let the rounding functions $\lceil x \rceil : \mathbb{R} \to \mathcal{F}$ and $\lfloor x \rfloor : \mathbb{R} \to \mathcal{F}$ be such that $\lfloor x \rfloor = \max\{y \in \mathcal{F} \,|\, y \leq x\}$ and $\lceil x \rceil = \min\{y \in \mathcal{F} \,|\, y \geq x\}$, which return the previous and next representable floating point numbers to $x$. Probabilistic Rounding can be defined as:

$$\circ_{\mathrm{PR}}(x, p_\circ) = \begin{cases} \lfloor x \rfloor & \text{with probability } p_\circ \\ \lceil x \rceil & \text{with probability } 1 - p_\circ \end{cases} \tag{2}$$

where $p_\circ : \mathbb{R} \to [0, 1]$.

The existing stochastic arithmetic techniques can then be reinterpreted with the PR definition.

### A.2.1   CESTAC Rounding

The Contrôle et Estimation STochastique des Arrondis de Calculs (CESTAC) technique simulates the round-off error in floating-point arithmetic by rounding upward or downward the result of each floating-point operation with equal probabilities.

$$\circ_{\mathrm{CESTAC}}(x) = \begin{cases} \lfloor x \rfloor & \text{with probability } \frac{1}{2} \\ \lceil x \rceil & \text{with probability } \frac{1}{2} \end{cases} \tag{3}$$

CESTAC is implemented in the CADNA library (Jézéquel & Chesneaux, 2008).

### A.2.2 Stochastic Rounding

The Stochastic Rounding (Forsythe, 1959) (SR) technique rounds the result of each floating-point with a probability that depends on the distance between the exact value and the two closest representable floating-point numbers. The probability is computed as:

$$\circ_{\text{SR}}(x) = \begin{cases} \lfloor x \rfloor & \text{with probability } p_{\text{SR}} \\ \lceil x \rceil & \text{with probability } 1 - p_{\text{SR}} \end{cases} \quad (4)$$

where $p_{\text{SR}}(x)$ is defined as:

$$p_{\text{SR}}(x) = 1 - \frac{x - \lfloor x \rfloor}{\lceil x \rceil - \lfloor x \rfloor}$$

SR implementations include, but are not limited to, the Random Rounding (RR) mode of MCA in Verificarlo, the implementation of Fasi and Mikaitis and the average rounding mode of Verrou.

## B   NAS Parallel Benchmarks description

| Benchmark | Description |
|-----------|-------------|
| bt | Block Triangular Solver |
| cg | Conjugate Gradient |
| ep | Embarrassingly Parallel |
| ft | Fast Fourier Transform |
| lu | Lower-Upper Symmetric Gauss-Seidel |
| mg | Multi-Grid Solver |
| sp | Scalar Pentadiagonal Solver |

Table 1: NAS Parallel Benchmarks description

## C   Deep Learning Use Cases

We evaluated the accuracy and performance of Fuzzy PyTorch across three deep-learning models demonstrating real-world applicability —MNIST (handwritten digit classification), WavLM (applied to speech-based Parkinson's disease identification), and FastSurfer (brain segmentation using CNN).

**MNIST**   To evaluate the performance and behavior of Fuzzy PyTorch, we conducted experiments on a small CNN trained on the MNIST dataset (LeCun, 1998). The model architecture consisted of convolutional, ReLU, max-pooling, convolutional, ReLU, dropout, flatten, linear, ReLU, dropout, and linear layers, followed by a log-softmax output layer. The task, a classification problem to identify digits from the input images, is well-established and widely considered solved. This experiment served as a baseline to demonstrate the feasibility and potential benefits of Fuzzy PyTorch in a controlled, well-understood context. In order to quantify the numerical variability in MNIST, we use the significant bit metric (Parker, 1997; Sohier et al., 2021), which calculates the amount of shared information among the perturbed results; the more significant bits, the greater the precision. We estimate the number of significant bits using the non-parametric method described in (Sohier et al., 2021), which is implemented in the `significant_digits` package (Verificarlo, 2024).

**WavLM**   The model for Parkinson's identification from speech is a pipeline composed of the WavLM Large (Chen et al., 2022) model in a frozen configuration to extract features from the audio recordings that are fed to the Emphasized Channel Attention, Propagation, and Aggregation Time Delay Neural Network (ECAPA-TDNN) (Desplanques et al., 2020) and a linear layer classifier followed by a max operation to obtain binary classification to determine whether a subject has Parkinson's disease or is a healthy control

subject. The model was trained on the mPower speech dataset (Bot et al., 2016). We will refer to this pipeline as WavLM in this work. For this use case, we also use significant bits to quantify the model's numerical variability.

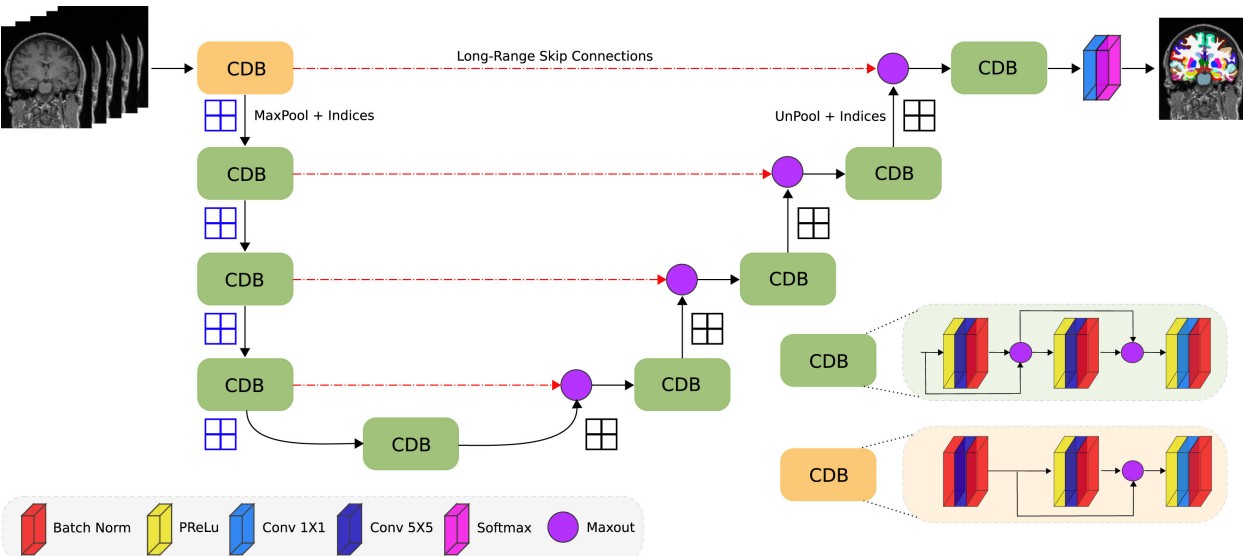

Figure 8: Illustration of FastSurfer's architecture. The CNN consists of four competitive dense blocks (CDB) in the encoder and decoder part, separated by a bottleneck layer. Figure reproduced from (Henschel et al., 2020).

**FastSurfer** FastSurfer (Henschel et al., 2020) is a CNN model that performs whole-brain segmentation, cortical surface reconstruction, fast spherical mapping, and cortical thickness analysis from anatomical MRI. The FastSurfer CNN is inspired by the QuickNAT model (Roy et al., 2019), which is composed of three 2D fully convolutional neural networks—each associated with a different 2D slice orientation—that each have the same encoder/decoder U-net architecture with skip connections, unpooling layers and dense connections as QuickNAT. A diagram of the model's architecture is available in the Appendix Figure 8. We focus exclusively on the task of whole-brain segmentation, defined as voxel-wise anatomical labeling of brain regions. This segmentation step is entirely performed by the CNN, without surface reconstruction, and uses the pre-trained FastSurfer model (v2.1.0) available on GitHub (Deep-MI, 2024). FastSurfer has demonstrated high accuracy, strong generalization to unseen datasets, and high test-retest reliability. This model serves as an ideal benchmark for studying numerical variability in high-dimensional medical imaging tasks due to its scientific relevance and architectural complexity. In our experiments, we applied FastSurfer to segment five subjects from the CoRR dataset (Zuo et al., 2014). By comparing Fuzzy PyTorch and Verrou instrumentation on FastSurfer inference, we aim to partially replicate previous findings on numerical uncertainty in FastSurfer (Gonzalez-Pepe et al., 2023), thereby validating the accuracy, reliability, and applicability of Fuzzy PyTorch to realistic large-scale segmentation tasks. For FastSurfer, we cannot compute significant bits as its segmentations are composed of categorical integer labels. Therefore, we compute the minimum Sørensen-Dice score across pairs of stochastic arithmetic runs. The Sørensen-Dice score measures the overlap between two segmentation results and is commonly used to quantify similarity between labeled regions in medical imaging. Further details on the Sørensen-Dice score are provided in Appendix D.

## D Sørensen-Dice Scores

The neuroimaging use case, the FastSurfer convolutional neural network (CNN), produces brain segmentations from anatomical Magnetic Resonance Images (MRI), which label different brain regions. In order to evaluate brain segmentations, we cannot use the significant bits metric. Segmentation tools like FastSurfer

generate categorical variables encoded as integers to represent segmentation labels, even though they rely on floating-point operations. Therefore, the significant bits metric cannot be applied as it is only useful for programs that produce floating-point outputs.

To assess the impact of numerical perturbations on segmentation stability, we compute the minimum Sørensen-Dice score across pairs of stochastic arithmetic runs. The Sørensen-Dice score measures the overlap between two segmentation results and is commonly used to quantify similarity between labeled regions in medical imaging. For each subject, we run FastSurfer multiple times with SA-enabled perturbations, producing $N$ different segmentations. Each segmentation output assigns a class label to every voxel in the brain MRI. Let $S_i$ and $S_j$ represent the set of voxels assigned to a specific brain region in the $i$-th and $j$-th stochastic arithmetic iterations, respectively. The Sørensen-Dice score between these two segmentations is given by:

$$\text{min Sørensen-Dice Score} = \min_{\substack{i,j \in \{1,\dots,N\} \\ i \neq j}} \left( \frac{2 \cdot |S_i \cap S_j|}{|S_i| + |S_j|} \right)$$

where $|S_i \cap S_j|$ is the number of overlapping voxels classified as part of the same region in both segmentations, and $|S_i|$ and $|S_j|$ are the total number of voxels assigned to that region in each iteration.

## E  Statistical Analysis of Harmonic Series Variance Homogeneity

| Descriptive Statistics by Method | | | | |
|---|---|---|---|---|
| **Method** | **Mean Std. Dev.** | **95% CI** | **F-statistic vs. PRISM SR** | **p-value** |
| PRISM SR | $1.65 \times 10^{-4}$ | $[1.08, 2.23] \times 10^{-4}$ | — | — |
| MCA RR | $2.41 \times 10^{-4}$ | $[1.37, 3.45] \times 10^{-4}$ | $F = 1.58$ | $p = 0.210$ |
| Verrou SR | $1.48 \times 10^{-4}$ | $[1.00, 1.96] \times 10^{-4}$ | $F = 0.21$ | $p = 0.651$ |
| FM SR | $2.10 \times 10^{-4}$ | $[1.27, 2.94] \times 10^{-4}$ | $F = 0.78$ | $p = 0.380$ |

Table 2: Statistical analysis of variance homogeneity across stochastic rounding methods in harmonic series computation. Levene's test confirms homogeneity of variances across stochastic rounding implementations ($F = 1.26$, $p = 0.29$), supporting the validity of comparative analyses. All pairwise F-tests compare against PRISM SR as the reference method. Tests performed at $\alpha = 0.05$ significance level.

