# OpenReview forum: "Fuzzy PyTorch: Rapid Numerical Variability Evaluation for Deep Learning Models"
_TMLR — Accepted by TMLR_

### Review · Reviewer_gaX7 · 2025-12-01

**Summary Of Contributions:**

In this article, the authors introduce Fuzzy PyTorch, a framework enabling large-scale, efficient stochastic arithmetic (SA) instrumentation for deep-learning models through PyTorch. The work provides a full-stack system for perturbing floating-point operations in PyTorch, enabling fast evaluation of numerical variability. Experiments on the harmonic series, NAS Parallel Benchmarks, MNIST, WavLM, and FastSurfer illustrate the performance gains obtained by the new framework.

**Additional Comments:**

None

**Audience:**

Yes

**Audience Explanation:**

The topic of this article is clearly in the scope of the journal.

**Claims And Evidence:**

Yes

**Claims Explanation:**

The paper identifies an increasingly critical issue, namely numerical variability in deep learning, which is receiving recently renewed attention due to mixed-precision hardware, accelerators (AWS Trainium, AMD MI300, NVIDIA Blackwell), and scientific use cases. The motivation is solid, and the authors correctly position their work within this broader context.
The PRISM library is convincingly presented as a performance-oriented alternative to existing SA runtimes.

**Requested Changes:**

Here are my comments and requested changes :
- Add an outline of the article at the end of the introduction. It should briefly summarize the structure (Sections 2–6 + Appendix).
- Add a short introductory paragraph at the beginning of Section 2, similar to the introduction you already provide at the start of Section 3.
- The introduction of the new term probabilistic rounding is unclear. Could you please justify why this new term is necessary rather than using the established term stochastic arithmetic (SA). Alternatively, consider removing “probabilistic rounding” and using SA consistently, unless a strong motivation is provided.
- IEEE-754 rounding (2.1.1) is not a probabilistic rounding method. It could be better to move it before Section 2.1, or introduce a short transitional subsection distinguishing deterministic and probabilistic rounding.
- My main concern is about the Up-Down Rounding. Indeed, I don't really understand the interest of this method compared to IEEE-754. Could you please extend this point ? Can it be seen as a stochastic version of IEEE-754 in order to add variability and robustness ? Could you also developp the definition of $\epsilon$ please ?
- Clarify in Section 3.1 that PRISM currently supports SR and UD only, and explicitly state whether support for CESTAC is planned or out of scope.
- Several figures, especially Figure 2, are difficult to interpret. Please revise them with more distinct shapes/colors, possibly larger markers or thicker lines and more detailed commentary in the text (particularly for Figure 2).

---

> ### Author Response · Authors · 2025-12-18
> **Response**
>
> We thank the reviewer for their detailed feedback on presentation and methodology and have integrated it into the revised paper. We address each concern below.
>
>
> **1. Paper Structure and Section Introductions: Add an outline of the article at the end of the introduction. It should briefly summarize the structure (Sections 2–6 + Appendix). Add a short introductory paragraph at the beginning of Section 2, similar to the introduction you already provide at the start of Section 3.** \
> Our Response: \
> We have added: \
> Paper outline at the end of the introduction summarizing the structure. \
> Section 2 has been restructured to address comments from other reviewers. An introductory paragraph has been added matching the style of Section 3's introduction
>
> **2. Probabilistic Rounding Terminology: The introduction of the new term probabilistic rounding is unclear. Could you please justify why this new term is necessary rather than using the established term stochastic arithmetic (SA). Alternatively, consider removing “probabilistic rounding” and using SA consistently, unless a strong motivation is provided.** \
> Our Response: \
> We follow the terminology established by Higham et al. (https://epubs.siam.org/doi/abs/10.1137/18M1226312), which uses "probabilistic rounding" as an umbrella term for methods where the rounding target (floor or ceiling) is determined probabilistically. This includes CESTAC, stochastic rounding, and up-down rounding—all unifying under how the rounding probability changes, not the rounding operation itself.
> We do not use "stochastic arithmetic" because it encompasses broader techniques beyond rounding modes (e.g., perturbations, error accumulation), whereas our discussion focuses specifically on rounding strategies.
>
> **3. IEEE-754 vs. Probabilistic Methods Organization:IEEE-754 rounding (2.1.1) is not a probabilistic rounding method. It could be better to move it before Section 2.1, or introduce a short transitional subsection distinguishing deterministic and probabilistic rounding.** \
> Our Response: \
> We have restructured this part of the paper to address comments from other reviewers. The previous section on probabilistic methods has been moved to the Appendix, with the exception of up-down rounding (our contribution). We also reorganized this section to introduce rounding methods conceptually and clearly distinguish deterministic (IEEE-754) from probabilistic methods (CESTAC, SR) as well as put IEEE rounding in its own section.
>
> **4. Up-Down Rounding: My main concern is about the Up-Down Rounding. Indeed, I don't really understand the interest of this method compared to IEEE-754. Could you please extend this point ? Can it be seen as a stochastic version of IEEE-754 in order to add variability and robustness ? Could you also develop the definition of  epsilon please?** \
> Our Response: \
> We aim to introduce UD rounding as a rounding mode that theoretically achieves faster variability estimation compared to stochastic rounding (SR). IEEE-754 rounding is a deterministic mode and therefore does not introduce numerical variability by design. In contrast, UD rounding is a probabilistic technique intended to deliberately inject stochasticity into floating-point operations in order to expose and measure numerical uncertainty. We include IEEE-754 as a baseline because it represents the default and most widely used rounding mode in practice, while UD serves as a simple stochastic counterpart that enables variability analysis rather than robustness by itself. However, experiments show that its practical performance in large-scale DL experiments is more limited than expected, because SR's exact operation preservation skips redundant computations in dense transformer layers while UD performs these operations. We recognize the confusion in the original text, and have revised all mentions of UD to emphasize that while it is the fastest rounding mode in small numerical experiments, its speedup ultimately does not scale.
> Epsilon (ε) definition: In Section 2.1, we have explicitly defined epsilon as the unit in the last place if x != 0 and 0 otherwise.
>
> **5. PRISM Backend Support Clarification: Clarify in Section 3.1 that PRISM currently supports SR and UD only, and explicitly state whether support for CESTAC is planned or out of scope.** \
> Our Response: \
> We have added explicit clarification in the Implementation section: PRISM currently only supports stochastic rounding (SR) and up-down rounding (UD), but not  CESTAC. CESTAC is less commonly used and therefore we do not currently support it, but we consider it to be future work.
>
> **6. Figure Legibility: Several figures, especially Figure 2, are difficult to interpret. Please revise them with more distinct shapes/colors, possibly larger markers or thicker lines and more detailed commentary in the text (particularly for Figure 2).** \
> Our Response: \
> We have revised our figures to be clearer, particularly for improved rendering at publication quality.

---

### Review · Reviewer_58Bm · 2025-12-01

**Summary Of Contributions:**

This paper presents Fuzzy PyTorch, a new package for assessing floating-point precision in PyTorch. The package integrates with PyTorch and can be used easily without modifying the original codebase. Numerical experiments on various benchmark datasets demonstrate that the package delivers significant speedups compared to existing packages.

**Additional Comments:**

N.A.

**Audience:**

Yes

**Audience Explanation:**

The software implementation is likely a good contribution to the community.

**Broader Impact Concerns:**

N.A.

**Claims And Evidence:**

No

**Claims Explanation:**

**Weaknesses**

Although the package itself can be a valuable contribution, I find the paper not sufficiently well written. I also have some concerns regarding the application of the package in the deep learning context.

1. Organization of the results and repetitive contents

   The organization of the contents sometimes looks redundant and confusing. For example,

   - The paragraphs before the contribution section jump back and forth between the features of Fuzzy PyTorch and the existing packages.

   - **Section 2.1** simply enumerates the existing rounding approaches. Since the package only implements two of them, I think they can be placed in the appendix.

     Similarly, **Section 4.1** enumerates some existing packages and **Section 4.2** enumerates the descriptions of the benchmark problems. The details can also be put in the appendix.

   - It seems **Section 2.3** should be part of the experiment setup.

   There are also a number of typos/inconsistencies throughout the paper (see **Minor issues**).

2. Compatibility issues when reproducing the experiments

   When I try to reproduce the experiments in the paper, I notice several compatibility issues (see **reproducibility issues** below. Moreover, the project's GitHub does not contain sufficient troubleshooting details.

3. Missing documentation

   The project's GitHub repo only tells how to reproduce the experiment. It does not provide documentation or examples for using the package.

4. Missing support for GPU floating-point assessment

   Currently, the package is written to run on a CPU. Although the authors mention that "the main conclusions from CPU testing are expected to generalize to GPU architectures," most deep learning training tasks are actually done on GPU. Therefore, I believe having GPU support is necessary for the package to be truly useful to the community. I understand that it is unlikely to add GPU support in the near term, but this should be an important future step for this project.

Overall, I believe that the package has the potential to become a useful tool for deep learning. However, the paper is not well written, and the package still has some compatibility issues. Please carefully address the first three weaknesses in the revision.

**Questions**

1. Could you please add troubleshooting instructions and documentations to your github repository?
2. How much extra work do you think is required for adding GPU support?
3. In **Section 3.3**, multiple PyTorch and Python versions are mentioned. Which versions does your package actually support?

**Minor issues**

Please carefully proofread the paper and fix typos/inconsistencies.

1. Page 1

   up-down and Up-Down rounding are used inconsistently.

2. Page 1

   Why are some references (Anonymized)?

3. Page 2

   "SR preserves exact values" is unclear.

4. Page 2

   In the contributions, FastSUfer and WavLM should have corresponding references when they first appear.

5. Page 3

   The introductory paragraph for the first component is too long.

6. Page 3

   $p$ the precision => $p$ is the precision.

7. Page 3

   $\mathbb{R} \rightarrow [0, 1]$ misses "."

8. Page 3

   $\mathbb{F}$ => $\mathcal{F}$

9. Page 3

   "is the most widely adopted deterministic rounding strategy defined by the IEEE-754 standard" seems unclear. Does IEEE-754 define multiple rounding strategies?

10. Page 3

    CESTAC means "Controle et Estimation STochastique des Arrondis de Calculs"

11. Page 4

    You might use UD directly after it's defined.

12. Page 4

    "the result of each floating-point operation rounded with..." is unclear.

13. Page 6

    "input regions such as zero" is unclear.

14. Page 8

    class "S" is not defined.

**Reproducibility issues**

I encountered the following issues when running the experiments on a Mac mini with an Apple M1 chip and macOS 15.7.2 (24G325). It seems that the package assumes a linux environment has some compatibility issues on arm64 architecture. I would suggest the authors explicitly mention the supported platform/architecture

1. When I run `./build.sh` directly, I get the following error

```
30.33 Need to get 0 B/5991 kB of archives.
30.33 After this operation, 0 B of additional disk space will be used.
30.33 Get:1 /bazelisk-amd64.deb bazelisk amd64 1.26.0 [5991 kB]
30.39 debconf: delaying package configuration, since apt-utils is not installed
30.41 dpkg: error processing archive /bazelisk-amd64.deb (--unpack):
30.41  package architecture (amd64) does not match system (arm64)
30.41 Errors were encountered while processing:
30.41  /bazelisk-amd64.deb
30.42 E: Sub-process /usr/bin/dpkg returned an error code (1)
------
Dockerfile-tools:3
--------------------
   2 |
   3 | >>> RUN apt-get update && \
   4 | >>>   apt-get install -y --no-install-recommends build-essential git wget curl bc \
   5 | >>>   automake libtool parallel \
   6 | >>>   clang-15 libclang-15-dev llvm-15 llvm-15-dev llvm-15-tools \
   7 | >>>   libomp-15-dev libomp5-15 \
   8 | >>>   libgmp-dev libgmpxx4ldbl libmpfr-dev libmpfr6 \
   9 | >>>   python3-pip && \
  10 | >>>   wget --no-check-certificate https://github.com/bazelbuild/bazelisk/releases/download/v1.26.0/bazelisk-amd64.deb && \
  11 | >>>   apt install -y  ./bazelisk-amd64.deb && \
  12 | >>>   apt-get clean && \
  13 | >>>   rm -rf /var/lib/apt/lists/*
  14 |
--------------------
ERROR: failed to build: failed to solve: process "/bin/sh -c apt-get update &&   apt-get install -y --no-install-recommends build-essential git wget curl bc   automake libtool parallel   clang-15 libclang-15-dev llvm-15 llvm-15-dev llvm-15-tools   libomp-15-dev libomp5-15   libgmp-dev libgmpxx4ldbl libmpfr-dev libmpfr6   python3-pip &&   wget --no-check-certificate https://github.com/bazelbuild/bazelisk/releases/download/v1.26.0/bazelisk-amd64.deb &&   apt install -y  ./bazelisk-amd64.deb &&   apt-get clean &&   rm -rf /var/lib/apt/lists/*" did not complete successfully: exit code: 100
```

The `install.sh` script also seems to have some compatibility issues

```
22.05 error: unknown FP unit 'sse'
22.06 make: *** [Makefile:38: stochrndhp.o] Error 1
22.06 make: Leaving directory '/build/tools/stochastic-rounding-evaluation/performance'
------
Dockerfile-tools:30
--------------------
  29 |
  30 | >>> RUN chmod +x /build/tools/install.sh && \
  31 | >>>   cd /build/tools && \
  32 | >>>   /build/tools/install.sh
--------------------

```

**Requested Changes:**

See weaknesses and minor issues.

---

> ### Author Response · Authors · 2025-12-18
> **Response Part 1**
>
> We thank the reviewer for their detailed feedback on the paper and the code and have integrated it into the revised paper. We address each concern below. Below, we address each concern directly in a two part response.
>
>
> **1. Organization of the results and repetitive contents: The organization of the contents sometimes looks redundant and confusing. For example, the paragraphs before the contribution section jump back and forth between the features of Fuzzy PyTorch and the existing packages. Section 2.1 simply enumerates the existing rounding approaches. Since the package only implements two of them, I think they can be placed in the appendix. Similarly, Section 4.1 enumerates some existing packages and Section 4.2 enumerates the descriptions of the benchmark problems. The details can also be put in the appendix. It seems Section 2.3 should be part of the experiment setup.** \
> Our Response: \
> We have reorganized the manuscript into the following structure: Introduction, Fuzzy Pytorch Design and Implementation (covering our up–down rounding contribution, the PRISM backend, and the Fuzzy PyTorch integration), Results, and Conclusion. Existing rounding modes are now consolidated in Appendix A, and the description of baseline tools (formerly Section 4.1) has been streamlined. We have also removed the standalone sections on uncertainty-quantification metrics; these are now referenced directly within the Experiments section or redirected to the Appendix or the original sources as appropriate.
>
>
> **2. There are also a number of typos/inconsistencies throughout the paper (see Minor issues).** \
> Our Response: \
> We have addressed all identified minor issues and typographical errors. Furthermore, we anonymized references involving authors of this work to preserve anonymity during review.
>
>
> **3. Platform and Architecture Support / Reproducibility Issues: Compatibility issues when reproducing the experiments. When I try to reproduce the experiments in the paper, I notice several compatibility issues (see reproducibility issues below. Moreover, the project's GitHub does not contain sufficient troubleshooting details. I encountered the following issues when running the experiments on a Mac mini with an Apple M1 chip and macOS 15.7.2 (24G325). It seems that the package assumes a linux environment has some compatibility issues on arm64 architecture. I would suggest the authors explicitly mention the supported platform/architecture** \
> Our Response: \
> We have added explicit statements in our Github documentation that Fuzzy PyTorch was designed for compatibility with Linux systems, because Verificarlo and PRISM were primarily designed for Linux, so our tool inherits its limitations. It should support ARM architecture if compiled locally.

---

> ### Author Response · Authors · 2025-12-18
> **Response Part 2**
>
> **4. Missing documentation: The project's GitHub repo only tells how to reproduce the experiment. It does not provide documentation or examples for using the package. Could you please add troubleshooting instructions and documentations to your github repository?** \
> Our Response: \
> We have revised the GitHub documentation by expanding the quickstart guide in the README.md. The user can now access additional documentation on how to run/set-up Verificarlo and the PRISM backend, which are essential components of Fuzzy PyTorch and setting up their custom use cases.
>
>
> **5. GPU Support Limitations: Missing support for GPU floating-point assessment
> Currently, the package is written to run on a CPU. Although the authors mention that "the main conclusions from CPU testing are expected to generalize to GPU architectures," most deep learning training tasks are actually done on GPU. Therefore, I believe having GPU support is necessary for the package to be truly useful to the community. I understand that it is unlikely to add GPU support in the near term, but this should be an important future step for this project. How much extra work do you think is required for adding GPU support?** \
> Our Response: \
> GPU support for stochastic arithmetic is not yet available in Fuzzy PyTorch as it is not supported in the underlying tools as well as in other comparable frameworks such as Verrou.
> Extending these methods to GPUs is a major undertaking that requires specialized approaches for different vendor platforms and substantial engineering effort—some of which we outlined in our conclusion. While this limitation is important to acknowledge, it also underscores the value of our contribution: providing a significantly faster CPU-based solution at a time when no practical GPU alternative exists. We revised our conclusion to highlight this contribution and to emphasize that developing GPU support is a key direction for future work.
>
> **6. Version Compatibility Clarification: In Section 3.3, multiple PyTorch and Python versions are mentioned. Which versions does your package actually support?** \
> Our Response: \
> We apologize for the confusion. We conducted all the experiments in this paper with PyTorch version 2.2.1, Verificarlo 2.0.0, LLVM version 7.0.0 and Python 3.8.5. Our statement “We have since also instrumented PyTorch version 2.6.0 with Python version 3.10 and LLVM 11.0.0 via Verificarlo 2.2.0.” was meant to imply that we have also instrumented other PyTorch versions to ensure compatibility and keep up with some of the more recent updates. This statement in the paper has been revised for clarity.

---

### Review · Reviewer_xFR1 · 2025-12-08

**Summary Of Contributions:**

The paper introduces the framework “Fuzzy Pytorch” for the evaluation of numerical variability in DL models. As DL models rely on computations sensitive to floating-point precision, assessing numerical variability is essential to ensure confidence in empirical results. The authors employ Stochastic Arithmetic to introduce controlled random perturbations into floating-point operations. This way, they enable statistical estimation of numerical variability. The proposed approach is practical as it requires no model modifications.

To this end, the authors introduce the PRISM library, which provides two rounding mechanisms: 1) stochastic rounding based on proximity to the representable floating-point number, and 2) up-down rounding. The benefits of PRISM include its easy applicability to existing PyTorch models and parallel execution capabilities, which enhance scalability. PRISM is evaluated across diverse use cases, including simple digit classification, segmentation, and speech classification.

**Additional Comments:**

There may be a problem with anonymity, as the GitHub link in the manuscript clearly reveals the lab that implemented the library. I would like to bring this issue to the attention of the AE.

**Audience:**

Yes

**Audience Explanation:**

I believe that the problem of numerical variability is an interesting and important topic for the DL community.

**Claims And Evidence:**

No

**Claims Explanation:**

- **Criticism of motivation and necessity:** My biggest concern with the paper is the problematic  motivation for the proposed framework. It is not sufficiently clear to me why a faster evaluation of numerical variability in DL models is necessary, especially given existing frameworks. The shortcomings that Fuzzy-PyTorch is supposed to fix are unclear to me.
The authors’ motivation is primarily based on the increased evaluation speed. Yet, if evaluation is performed only once, why is speed so critical? Are other frameworks considered infeasible in certain contexts? As long as this remains resolved and the subsequent results lack the necessary basis for relevance.
- **Unclear rationale/necessity of up-down (UD) rounding:** Another important point is the introduction of UD rounding. There is really no real justification for why this specific type of rounding is necessary.  What are the specific problems it solves in existing approaches? It remains unclear whether UD rounding was introduced solely to reduce computation time. Assuming the main argument is speed, this is inconsistent with experiments, as UD rounding does not offer speed-ups in transformer architectures. On the other hand, there is no complexity analysis that would prove a general speed advantage. Without a clear explanation of the rationale, the introduction of this technique does not appear justified.
- **Lack of Recommendations:** The next point relates to the use of the Sørensen-Dice score. Although this is justified for the segmentation problem presented, the authors do not provide any recommendations for other use cases and tasks. I believe a framework should be modular in design and go beyond a specific use case. There is a lack of recommendations on how users can transfer the tool to new use cases.
- **Minor Problems:** The paper has shortcomings in terms of readability and structure. In particular, the introduction lacks a clear line of argumentation. This structural weakness directly contributes to the fact that the motivation behind the framework, which was criticized in my first point, remains unclear. Additionally, there are some issues with acronyms. In some cases, abbreviations are introduced without being used or defined multiple times.

**Requested Changes:**

Given the criticism in the section “Claims and evidence,” could you comment on each of the points mentioned? Ideally, the issues raised there should be taken into account in a revision.

---

> ### Author Response · Authors · 2025-12-18
> **Response Part 1**
>
> We thank the reviewer for their constructive feedback and have made the recommended changes in the text to improve the structure and readability as well as address specific concerns. Below, we address each concern directly in a two part response.
>
>
> **1. Criticism of motivation and necessity: My biggest concern with the paper is the problematic motivation for the proposed framework. It is not sufficiently clear to me why a faster evaluation of numerical variability in DL models is necessary, especially given existing frameworks. The shortcomings that Fuzzy-PyTorch is supposed to fix are unclear to me. The authors’ motivation is primarily based on the increased evaluation speed. Yet, if evaluation is performed only once, why is speed so critical? Are other frameworks considered infeasible in certain contexts? As long as this remains resolved and the subsequent results lack the necessary basis for relevance.** \
> Our Response: \
> A practical challenge in studying numerical variability is the need to run programs multiple times to obtain stable statistical estimates. Following Sohier et al. (https://dl.acm.org/doi/abs/10.1145/3432184), 10 samples grant only an 80% confidence level, thus underlining the need for multiple samples. Moreover, we observe that existing tools, particularly Verrou, one of the most accessible stochastic arithmetic frameworks for deep learning, introduce slowdowns of 10–1000× per run in practice on deep learning models with only a few million parameters. In such settings, collecting enough samples takes days or weeks even with parallelization. Scaling these analyses to modern large language models becomes effectively infeasible.
> For numerical stability research to keep pace with rapidly growing deep learning architectures, it is essential to develop tools that introduce minimal computational overhead. This motivates Fuzzy PyTorch's design: prioritizing speed and scalability so that numerical variability studies remain tractable even on increasingly large models. Our 5–60× speedups over Verrou (now Section 3.4) make numerical analysis feasible within practical computational budgets. \
> Textual improvements: We have revised the introduction and conclusion to emphasize that speed is necessary for feasibility.
>
>
> **2. Unclear rationale/necessity of up-down (UD) rounding: Another important point is the introduction of UD rounding. There is really no real justification for why this specific type of rounding is necessary. What are the specific problems it solves in existing approaches? It remains unclear whether UD rounding was introduced solely to reduce computation time. Assuming the main argument is speed, this is inconsistent with experiments, as UD rounding does not offer speed-ups in transformer architectures. On the other hand, there is no complexity analysis that would prove a general speed advantage. Without a clear explanation of the rationale, the introduction of this technique does not appear justified.** \
> Our Response: \
> We aim to introduce UD rounding as a rounding mode that theoretically achieves faster variability estimation compared to stochastic rounding (SR). However, its theoretical advantages do not carry over to large-scale deep learning experiments, because SR's exact operation preservation skips redundant computations in dense transformer layers while UD performs these operations. \
> Textual improvements: We recognize the confusion in the original text, and have revised all mentions of UD to emphasize that while it is the fastest rounding mode in small numerical experiments, its speedup ultimately does not scale.

---

> > ### Author Response · Authors · 2025-12-18
> > **Response Part 2**
> >
> > **3. Lack of Recommendations: The next point relates to the use of the Sørensen-Dice score. Although this is justified for the segmentation problem presented, the authors do not provide any recommendations for other use cases and tasks. I believe a framework should be modular in design and go beyond a specific use case. There is a lack of recommendations on how users can transfer the tool to new use cases.** \
> > Our Response: \
> > Fuzzy PyTorch is task-agnostic and can, in principle, be applied to any experiment built in PyTorch. We demonstrate this flexibility by evaluating the tool on a range of numerical and deep learning tasks, from small-scale numerical experiments (NAS Parallel Benchmarks and the harmonic series) to larger, diverse deep learning applications (MNIST digit classification, FastSurfer brain segmentation, and WavLM-based Parkinson’s prediction). Each use case requires its own evaluation metrics. For floating-point outputs, we recommend using the significant bits metric (previously Section 2.2, consult Sohier et al:https://dl.acm.org/doi/abs/10.1145/3432184 for more information), which provides a general and informative measure of numerical variability. For tasks producing non-floating-point outputs (e.g., categorical labels), task-specific metrics remain necessary; for example, Sørensen–Dice for segmentation and accuracy for classification. While we describe the Dice score in Section 2.3 (now the Appendix), we do not expand on other widely used metrics such as standard deviation, accuracy, or precision, as these are well-established within the field. \
> > Textual improvements: We have expanded our GitHub documentation to provide clearer guidance for users on how to apply Fuzzy PyTorch in practice, including pointers to Verificarlo and the PRISM backend for more advanced or customized workflows. Furthermore, we have restructured our introduction and conclusion to clarify the importance of Fuzzy PyTorch as well as how it can facilitate systematic, feasible numerical analysis in deep learning and how that resulting variability can be leveraged into applications.
> >
> >
> > **4. Minor Problems: The paper has shortcomings in terms of readability and structure. In particular, the introduction lacks a clear line of argumentation. This structural weakness directly contributes to the fact that the motivation behind the framework, which was criticized in my first point, remains unclear. Additionally, there are some issues with acronyms. In some cases, abbreviations are introduced without being used or defined multiple times.** \
> > Our Response: \
> > We have restructured the paper to improve readability and structure and paid special attention to the introduction/conclusion to clarify our motivation.
> >
> >
> > **5. Anonymity Concern: There may be a problem with anonymity, as the GitHub link in the manuscript clearly reveals the lab that implemented the library. I would like to bring this issue to the attention of the AE.** \
> > Our Response: \
> > Our apologies, we made the Github link public to comply with TMLR's reproducibility standards, which we consider key, considering our paper proposes a software tool. We have now recently discovered a service that allows one to share a Github repository anonymously, and we have thus made the necessary changes.

---

### Decision · Action_Editor_xnro · 2026-01-26

**Recommendation:** Accept as is

**Audience:**

Yes

**Audience Explanation:**

This paper introduces a software framework for evaluating the numerical variability of deep learning models.  This seems useful given e.g. the proliferation of mixed / low precision hardware and relevant to the community.

**Claims And Evidence:**

Yes

**Claims Explanation:**

The reviewers all found that the claims were supported by the evidence.  There were some initial issues with the quality of writing and one reviewer was initially unable to reproduce results when using the software due to versioning issues.  However, it seems that the authors addressed these concerns in their response.